# Secure Multiparty Computation for Synthetic Data Generation from Distributed Data

**Mayana Pereira**
AI for Good Research Lab
Microsoft
Redmond, WA, USA
`mayana.wanderley@microsoft.com`

**Sikha Pentyala**
School of Engineering and Technology
University of Washington
Tacoma, WA, USA
`sikha@uw.edu`

**Anderson Nascimento**
School of Engineering and Technology
University of Washington
Tacoma, WA, USA
`andclay@uw.edu`

**Rafael T. de Sousa Jr.**
Department of Electrical Engineering
Universidade de Brasilia
Brasilia, Brazil
`desousa@unb.br`

**Martine De Cock**
School of Engineering and Technology
University of Washington
Tacoma, WA, USA
`mdecock@uw.edu`

## Abstract

Legal and ethical restrictions on accessing relevant data inhibit data science research in critical domains such as health, finance, and education. Synthetic data generation algorithms with privacy guarantees are emerging as a paradigm to break this data logjam. Existing approaches, however, assume that the data holders supply their raw data to a trusted curator, who uses it as fuel for synthetic data generation. This severely limits the applicability, as much of the valuable data in the world is locked up in silos, controlled by entities who cannot show their data to each other or a central aggregator without raising privacy concerns. To overcome this roadblock, we propose the first solution in which data holders only share encrypted data for differentially private synthetic data generation. Data holders send shares to servers who perform Secure Multiparty Computation (MPC) computations while the original data stays encrypted. We instantiate this idea in an MPC protocol for the Multiplicative Weights with Exponential Mechanism (MWEM) algorithm to generate synthetic data based on real data originating from many data holders without reliance on a single point of failure.

## 1 Introduction

We live in an era of abundant data, where enormous amounts of personal data are collected daily via smartphones, social media, smartwatches, medical devices, among many other services. These datasets have helped researchers and industry understand our behavior better on both individual and collective levels, and have also allowed important research studies in many disciplines, including health, education, and economy. At the same time, we see an increase in privacy regulations globally.

NeurIPS 2022 Workshop on Synthetic Data for Empowering ML Research.

Following the introduction of the GDPR,[1] more than 60 jurisdictions around the world have proposed postmodern data privacy protection laws. By 2024, 75% of the world's population will have its personal information covered under modern privacy regulations [30]. While privacy regulations are of extreme importance from an ethics perspective, they can potentially result in data stored in silos, compromising data usage and data sharing, and stalling research.

Synthetic data generation is emerging as a paradigm to break this data logjam. While data synthesis is arguably best known as a means to create training examples for data hungry deep learning models [27], it is increasingly acknowledged and proposed as a privacy-enhancing technology (PET) [18, 22, 32, 33, 35, 36]. When done well, synthetic data has the same distribution or characteristics as the underlying, real data, but, crucially, without replicating personal information. The latter is often formalized through the notion of Differential Privacy (DP) [14], which intuitively means that the synthetic data should not reveal specifics about *individual* records in the underlying, real data.

What the existing approaches for generating synthetic data and publication of data with DP guarantees have in common, is that they all assume that the original, real data, exists with one data holder, or, if the data originates from different data holders, that the latter are able to send their data to a central aggregator who in turn will use it as input for synthetic data generation or DP publication algorithms. Much of the valuable data in the world however is under the control of entities (companies, banks, hospitals, biomedical research institutes etc.) who cannot show their data to each other or to a central aggregator without raising privacy concerns. This is the bottleneck that we address, namely *how to generate synthetic data based on the combined data from multiple data holders that no one is allowed to see*. This includes data that is horizontally distributed, such as healthcare data across different hospitals, or financial data held by different banks, as well as data that is vertically distributed, such as advertising data where the publishers hold the input features while advertisers have the label, and many more. Finally, in addition to the cross-silo scenarios described above, our proposed solution makes a scenario practical where millions of users could provide their data to produce a synthetic dataset in a way that private, individual data would never be exposed in plaintext (i.e. without being encrypted) to any entity – practically implementing a "synthetic data as a service" model.

The contributions of this paper are the following: (1) We introduce a framework for synthetic data generation from distributed databases that utilizes Secure Multiparty Computation (MPC) [10] protocols that are run by two or more computing parties to emulate a trusted curator. This simulation enables the generation of synthetic data from training data held by multiple data holders, without requiring these data holders to disclose their data to anyone in an unencrypted manner. (2) We modify the Multiplicative Weights with Exponential Mechanism (MWEM), to generate synthetic data with DP guarantees, based on real data originating from many data holders, and without reliance on a single point of failure. (3) We propose an MPC protocol for secure sampling from distributed data using the exponential mechanism.

**Novelty w.r.t. existing work.**  Previous proposals for differentially private synthetic data generation from distributed databases use federated learning (FL) for training the data synthesizer [4, 37, 38]. In these methods, each data holder sends model weights (without privacy protection) to a trusted aggregator, who computes the average of model weights and adds Laplacian noise. Our proposal removes the need for data holders to disclose model parameters, and the need to rely on a single point of failure, by emulating the trusted aggregator with MPC. Additionally, previous works utilizing FL to train data synthesizers only account for horizontally partitioned data.

While MPC has emerged as a paradigm for privacy-preserving training of ML models over distributed data (e.g. [1, 2, 13, 16, 26, 34]) and privacy-preserving inference with trained ML models (e.g. [12, 15, 21, 25, 29]), and it has been proposed for secure computation of histograms (e.g. [5]), the idea of using MPC for privacy-preserving generation of synthetic data, as we propose here, is novel and a practical and secure technological solution.

## 2   Preliminaries

**Differential Privacy** [14]. A randomized algorithm $\mathcal{M}$, which takes inputs from an input space $\mathcal{D}$ and outputs values from an output space $\mathcal{O}$, provides $\epsilon$-Differential Privacy if for all subsets $\mathcal{S} \subseteq \mathcal{O}$,

---

[1]European General Data Protection Regulation `https://gdpr-info.eu/`

**Algorithm 1:** The MWEM algorithm [17]

**Input:** Dataset $D$ over a universe $\mathcal{D}$, set of linear queries $Q$, number of iterations $T$, and privacy
      parameter $\epsilon > 0$.

Let $n$ denote $|D|$, the number of records in $D$.

Let $A_0$ denote $n$ times the uniform distribution over $\mathcal{D}$.

**1 for** $i \in \{1, \ldots, T\}$ **do**

**2**      **Exponential Mechanism:** sample a query $q_i \in Q$ using the Exponential Mechanism parametrized
        with epsilon value $\epsilon/2T$ and the score function: $s_i(D, q) = |q(A_{i-1}) - q(D)|$

**3**      **Laplace Mechanism:** Let measurement $m_i = q_i(D) + \mathsf{Lap}(2T/\epsilon)$

**4**      **Multiplicative Weights:** Let $A_i$ be $n$ times the distribution whose entries satisfy

$$A_i(x) \propto A_{i-1}(x) \times \exp(q_i(x) \times (m_i - q_i(A_{i-1}))/2n)$$

**Output:** $A = \mathrm{avg}_{i<T} A_i$

and for all neighboring databases $D$ and $D' \in (D)$ (i.e., $D$ and $D'$ differ in at most one entry),

$$\Pr[\mathcal{M}(D) \in \mathcal{S}] \leq e^\epsilon \cdot \Pr[\mathcal{M}(D') \in \mathcal{S}]$$

The DP concept quantifies privacy loss while providing an understandable notion of privacy: outputs of a DP algorithm for datasets that vary by a single entry are indistinguishable, bounded by the privacy parameter $\epsilon$. In this paper we use the well known exponential and Laplace mechanisms for creating $\epsilon$-DP algorithms (see App. A). We can relax this definition by adding an additive term $\delta$ to it: $\Pr[\mathcal{M}(D) \in \mathcal{S}] \leq e^\epsilon \cdot \Pr[\mathcal{M}(D') \in \mathcal{S}] + \delta$. This relaxed notion is called approximate differential privacy or $(\epsilon, \delta)-$DP.

**Multiplicative Weights with Exponential Mechanism Algorithm** [17]. The MWEM algorithm takes as input a dataset $D \subseteq \mathcal{D}$ and a set of linear queries $Q$ (e.g. counting queries).[2] The algorithm aims to produce a distribution $A$ over $\mathcal{D}$ such that the answers to the queries $q$ in $Q$ when run over $A$ are similar to when run over $D$, i.e. the difference between $q(A)$ and $q(D)$ should be small. This is achieved by repeatedly sampling a query for which the difference is still large (line 2 in Alg. 1), and updating the weight that $A$ places on each record $x$ with the Multiplicative Weights update rule to better approximate the distribution of $D$ w.r.t. $q$ (line 4). Furthermore, MWEM satisfies $\epsilon$-DP by leveraging the exponential mechanism for query selection, and the Laplace mechanism to perturb the query results.

**Secure Multiparty Computation (MPC).** MPC protocols enable a set of parties to jointly compute the output of a function over the private inputs of each party, without requiring any of the parties to disclose their own private inputs [9]. MPC protocols are designed to prevent and detect attacks by an adversary corrupting one or more parties to learn private information or to cause the result of the computation to be incorrect. The adversary – which we assume to be static – can have different levels of adversarial power. In the *semi-honest* model, even corrupted parties follow the instructions of the protocol, but the adversary attempts to learn private information from the internal state of the corrupted parties and the messages that they receive. MPC protocols that are secure against semi-honest or *"passive"* adversaries prevent such leakage of information. In the *malicious* adversarial model, the corrupted parties can arbitrarily deviate from the protocol specification. Providing security in the presence of malicious or *"active"* adversaries, i.e. ensuring that no such adversarial attack can succeed, comes at a higher computational cost than in the passive case. The protocols that we propose are sufficiently generic to be used in settings with passive or active adversaries. This is achieved by changing the underlying MPC scheme to align with the desired security setting. We consider an honest-majority 3-party computing setting out of which at most one party can be corrupted (3PC) [3, 11], an honest-majority 4-party computing setting with one corruption (4PC) [11], and a dishonest-majority 2-party computation setting where each party can only trust itself (2PC) [8]. In all these MPC schemes, data is encrypted by splitting it into secret shares, which are distributed to a set of computing parties that run MPC protocols and perform computations over these secret shares, in our case to generate synthetic data. As all computations are done over encrypted values, the servers do not learn the values of the inputs nor of intermediate results, i.e. MPC provides *input privacy*. Below we propose an MPC protocol to generate synthetic data under DP guarantees, i.e. providing

---

[2]A linear query $q$ is a function that maps data records in $\mathcal{D}$ to the interval $[-1, +1]$. By extension, the answer of a linear query $q$ on a dataset $D$ is defined as $q(D) = \sum_{x \in \mathcal{D}} q(x) \cdot D(x)$.

**Protocol 2:** $\pi_{\mathsf{QEM}}$ - Protocol for secure sampling a query using the Exponential Mechanism

**Input:** The number $N$ of queries in $Q$, secret-shared true query answer $[\![q_i(D)]\!]$ and approximate query
answer $q_i(A)$ for each $q_i$ in $Q$, privacy parameter $\epsilon' = \epsilon/(2T)$

1  Initialize a vector **err** of length $N$
2  **for** $i \leftarrow 1$ **to** $N$ **do**
3      $[\![\mathrm{diff}]\!] \leftarrow [\![q_i(D)]\!] - q_i(A)$
4      $[\![\mathrm{sign}]\!] \leftarrow \pi_{\mathsf{LT}}([\![\mathrm{diff}]\!], 0)$ // with secure comparison protocol $\pi_{\mathsf{LT}}$
5      $[\![\mathrm{abs\_diff}]\!] \leftarrow \pi_{\mathsf{MUL}}(1 - 2 \cdot [\![\mathrm{sign}]\!], [\![\mathrm{diff}]\!])$ //with secure multiplication protocol $\pi_{\mathsf{MUL}}$
6      $[\![err[i]]\!] \leftarrow [\![\mathrm{abs\_diff}]\!] \cdot 0.5 \cdot \epsilon'$
7  $[\![\mathrm{max\_err}]\!] \leftarrow \pi_{\mathsf{MAX}}([\![\mathbf{err}]\!])$ // with secure maximum protocol $\pi_{\mathsf{MAX}}$
8  **for** $i \leftarrow 1$ **to** $N$ **do**
9      $[\![err[i]]\!] \leftarrow \pi_{\mathsf{EXP}}([\![err[i]]\!] - [\![\mathrm{max\_err}]\!])$ // with secure exponentiation protocol $\pi_{\mathsf{EXP}}$
10  // Get random threshold to sample query
11  es $\leftarrow 0$
12  Initialize a vector **c** of length $N$
13  **for** $i \leftarrow 1$ **to** $N$ **do**
14      $[\![\mathrm{es}]\!] \leftarrow [\![\mathrm{es}]\!] + [\![err[i]]\!]$
15      $[\![c[i]]\!] \leftarrow [\![\mathrm{es}]\!]$
16  $[\![\mathrm{r}]\!] \leftarrow \pi_{\mathsf{GR-RANDOM}}(0, 1)$ // with protocol for random number generation $\pi_{\mathsf{GR-RANDOM}}$
17  $[\![\mathrm{t}]\!] \leftarrow \pi_{\mathsf{MUL}}([\![\mathrm{es}]\!], [\![\mathrm{r}]\!])$
18  $s \leftarrow 0$
19  **for** $i \leftarrow 1$ **to** $N$ **do**
20      $[\![\mathrm{cnd}]\!] \leftarrow \pi_{\mathsf{GT}}([\![c[i]]\!], [\![\mathrm{t}]\!])$
21      $[\![s]\!] \leftarrow [\![s]\!] + [\![\mathrm{cnd}]\!]$
22  $[\![\mathrm{cnd}]\!] \leftarrow \pi_{\mathsf{EQ}}([\![s]\!], 0)$
23  $[\![k]\!] \leftarrow N - \pi_{\mathsf{MUL}}([\![s]\!] - 1, 1 - [\![\mathrm{cnd}]\!])$
**Output:** Secret-sharing $[\![k]\!]$ of the index of the selected query

*output privacy* as well, so that the output of the synthetic data generation process can be published. We refer to App. B for more details on the MPC primitives used.

## 3  Method

We address the scenario where, instead of residing with one entity, the dataset $D$ that we wish to give as input to the MWEM algorithm is distributed among multiple data holders who cannot disclose their data to anyone in an unencrypted manner. We distinguish between the *data holders* who hold the data sets, and the *computing parties* who run the MPC protocols for synthetic data generation and noise addition. Our solution works in scenarios in which each data holder (e.g. hospital or bank) is also a computing party, as well as in scenarios where the data holders outsource the computations to untrusted servers (computing parties) instead. The data holders send secret shares of their data to a set of computing parties. Without loss of generality, we assume that the computing parties have secret shares of $[\![D]\!]$ of $D$, which they can use to compute a secret-sharing of the query result $[\![q(D)]\!]$ for each $q$ in $Q$ using primitive MPC protocols for addition and multiplication (see App. B).

The execution of the overall MWEM algorithm can be coordinated by one of the computing parties or data holders or any other entity interested in generating the synthetic data. Such a setup will not require the data holders to be online after they have secret shared the query answers based on their private data implementing data generation-as-a-service model which is independent of the number of data holders. Indeed, there are only two crucial steps in Alg. 1 that rely directly on the encrypted data $[\![D]\!]$, or rather $[\![q(D)]\!]$, hence requiring MPC computations involving all computing parties: (1) the query selection in line 2; and (2) the measurement in line 3. Note that the output of the computations in line 2 and line 3 is protected with DP guarantees. In other words, if we let the computing parties run MPC protocols for the computations and the DP mechanisms, then they can publicly reveal the selected query (line 2) and the perturbed query result (line 3), which can subsequently be used for further computations. Furthermore, there is no need to encrypt the synthetic data distribution $A$, as it is not based on any information from $D$ that is not already protected with DP. This is a welcome observation because it means that query evaluations need to be done only once over encrypted data,

---
**Protocol 3:** $\pi_{\text{LAP}}$ - Protocol for Laplace mechanism
---
**Input:** Secret shared true query answer $[\![q_i(D)]\!]$ and $b = 2T/\epsilon$

1  $[\![\text{x}]\!] \leftarrow \pi_{\text{GR-RANDOM}}(0, 1)$ // with protocol for random number generation $\pi_{\text{GR-RANDOM}}$
2  $[\![\text{ln\_x}]\!] \leftarrow \pi_{\text{LN}}([\![\text{x}]\!])$ // with secure logarithm protocol $\pi_{\text{LN}}$
3  $[\![\text{r}]\!] \leftarrow \pi_{\text{GR-RNDM-BIT}}()$ // with protocol for random bit generation $\pi_{\text{GR-RANDOM}}$
4  $[\![\text{c}]\!] \leftarrow 2 \cdot [\![\text{r}]\!] - 1$
5  $[\![m_i]\!] \leftarrow [\![q_i(D)]\!] + b \cdot \pi_{\text{MUL}}([\![\text{ln\_x}]\!], [\![\text{c}]\!])$ // with secure multiplication protocol $\pi_{\text{MUL}}$
**Output:** Secret sharing of measurement $[\![m_i]\!]$ for the query $q_i$, with $m_i = q_i(D) + \text{Lap}(2T/\epsilon)$
---

namely to compute $[\![q(D)]\!]$, and any further query evaluations on new versions of $A$ can be done in-the-clear, i.e. without the need for encryption.

**Description of $\pi_{\text{QEM}}$.** For the secure query sampling on line 2 of Alg. 1, we propose MPC-protocol $\pi_{\text{QEM}}$ (see Prot. 2) which is called with privacy budget $\epsilon' = \epsilon/(2T)$. $\pi_{\text{QEM}}$ consists of two parts: on lines 1–9 the parties compute secret shares of the probability distribution over the queries, while on lines 10–23 the parties subsequently sample a query $q_k$ from that distribution. Pseudocode for a corresponding algorithm in-the-clear, i.e. without regards for privacy, is given in Alg. 4 in App. A, while the MPC primitives used in Alg. 1 are described in App. B. The number and the kind of operations to construct the probability distribution and to compute the threshold (lines 1–9 in Alg. 4) are deterministic in the sense that they do not depend on the value of the data, hence their MPC counterpart in Prot. 2 is relatively straightforward. The implementation of the counterpart of the for-loop that starts on line 11 in Alg. 4 requires more care, as exiting the for-loop prematurely could allow an adversary to infer the value of the returned index from the runtime. The code in line 18-23 in Prot. 2 is written to prevent such side-channel attacks. To understand this part of the code, note that we have a list $c[1..N]$ of non-decreasing values, i.e. the cumulative probability sums, and we – or rather the computing parties – have to find the first index $i$ in $c[1..N]$ for which $c[i] > \text{t}$. In a mock example with $N = 10$, and assuming that the first such $c[i]$ value is at position 7, the tests on line 20 will generate the results 0,0,0,0,0,0,1,1,1,1. On line 21, these results are accumulated in $s$, which eventually becomes 4, and the desired index is computed as $N - (s - 1) = 10 - 3 = 7$. Lines 22–23 take care of the edge case when $c[i] \leq \text{t}$ for all $i$ (i.e. $s$ is 0). We protect the value of $s$ by employing MPC primitives for multiplication to simulate a conditional statement.

**Description of $\pi_{\text{LAP}}$.** For the measurement computed in line 3 of Alg. 1, we design $\pi_{\text{LAP}}$ (see Prot. 3) to securely sample noise from from the Laplacian distribution and add to the secret sharing of $q_i(D)$. The noise is sampled as $b \cdot \ln \text{x} \cdot \text{c}$ where $b = 2T/\epsilon$ is the privacy budget, x is a random value drawn from the uniform distribution in [0,1] and c is a random value selected from $\{-1, 1\}$. On lines 1–2, the parties straightforwardly compute x and its natural log. To compute c, the parties, on line 3, generate secret shares of a random bit $[\![\text{r}]\!]$, i.e. a value $\in \{0, 1\}$ is chosen, where each value has a chance of 50% to be chosen. On line 4, the parties transform r to a value $\in \{-1, 1\}$ using the logic c = 2$\cdot$ r $-1$. Line 5 is straightforward where the parties compute secret shares of measurement $[\![m_i]\!]$ for the query $q_i$, which is then made public for further computations in Alg. 1.

**Remark.** We provide a proof-of-concept of our proposed framework through the MWEM algorithm, a well-known algorithm for releasing data for statistical analysis. However, the proposed general framework can be adapted to other synthetic data generation techniques.

## 4   Experiments

We evaluate MPC-MWEM against a centralized version of MWEM [17] using two publicly available datasets, namely the Car and Adult datasets, which have been featured in previous DP synthetic dataset generation analyses [17, 31]. In all the results below, **centralized** refers to the setting in which all data holders disclose their data to a central, trusted curator who runs the MWEM algorithm over all the data combined, while **distributed** refers to the setting in which the data holders secret share their data with computing parties who run MPC protocols. The distributed setting protects the privacy of the inputs, while the centralized setting does not. The results for the centralized setting are obtained with an implementation of MWEM in SmartNoise [31]. For the distributed setting, we implemented our MPC protocols $\pi_{\text{QEM}}$ and $\pi_{\text{LAP}}$ in the MPC framework MP-SPDZ [19].[3]

---
[3]We will make the code available with the full version of the paper.

**Experimental Settings.** We empirically validate the utility of the produced synthetic data and measure performance by training logistic regression (LR) models using synthetic data and testing the models on real data, as in [31]. We evaluate model performance using AUC-ROC. We compare the performance of models trained on synthetic data generated in the centralized mode, and synthetic data generated in the distributed mode using MPC where the data is split horizontally across data holders. We repeat the comparison process for different privacy parameter values. We measure runtimes of our method for different numbers of MWEM iterations $T$ and compare with the centralized setting, while keeping other parameters constant.[4] We use a maximum number of iterations of 1000 and other default parameters of LR available in Scikit-learn [28] to train the models.

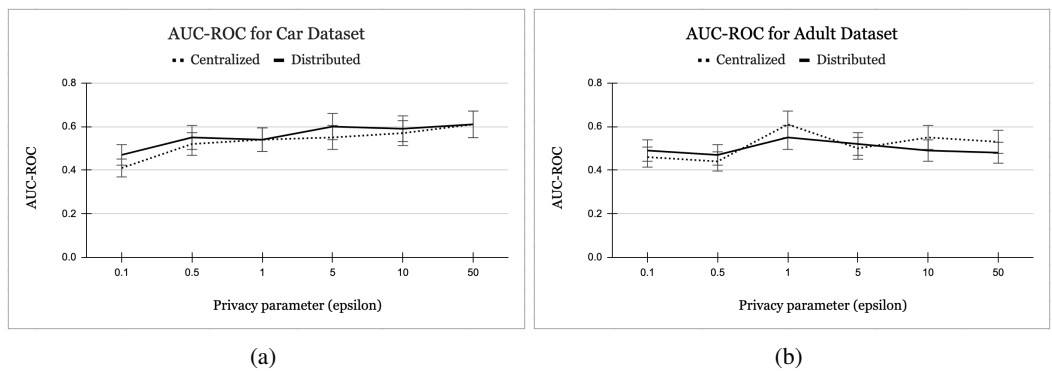

Figure 1: AUC-ROC of LR models trained on synthetic data generated by two different modes (centralized and distributed) with varying privacy budget. The results presented are averaged over 10 runs.

**Quantitative Analysis of Utility.** In Fig. 1, we investigate the trade-off between the privacy parameter $\epsilon$ and utility of models trained with synthetic data generated by the two different modes (centralized and distributed). The models perform similarly in terms of AUC-ROC, for the different values of $\epsilon$. Additionally, the trends are also consistent for both datasets. In the experiments using the Car dataset we see a upward trend for both modes, whereas for the adult dataset we see a small spike for $\epsilon = 1$ for both modes. Based on similar trendlines for both settings, we conclude that MPC emulates the centralized mode of operation. The small differences observed in the plots are a result of the noise introduced by the DP mechanisms. The results in Fig. 1 are averaged over 10 runs.

**Quantitative Analysis of Iterations and Runtime.** We measure runtime for different values of the number of iterations $T$, which is a hyperparameter of MWEM. Previous works have demonstrated the trade-off between the number of iterations and quality of the synthetic data [17]. Tab. 1 shows the runtime for different choices of $T$ averaged over 3 runs for the centralized setting and for the distributed setting with 2, 3, and 4 computing parties. All MPC based computations were done in ring $\mathbb{Z}_q$ with $q = 2^{64}$. As observed, the runtimes increase with $T$. We note that the runtimes further depend on the dimensions of the datasets and the number of queries, as shown in [17]. The increased runtimes for the distributed setting when compared to their corresponding centralized setting are due to the runtimes of the MPC protocols. For example, in a 3PC passive security setting for $|T| = 1$, each call to $\pi_{\mathsf{QEM}}$ adds $\sim$0.74 secs for $|Q| = 400$ and $\pi_{\mathsf{LAP}}$ adds $\sim$0.006 secs to the synthetic generation process. The differences in runtime observed across different security settings are in line with existing literature [11]. All the experiments were run on Azure D8ads_v5 8 vCPUs, 32Gib RAM. The runtimes of our proposed method depend only on the computing parties and the threat model. Our proposed MPC protocols can be made scalable and efficient by replacing the appropriate and efficient underlying MPC schemes.

We note that our approach works for any partitioning of data – horizontal, vertical or mixed – with appropriate MPC protocols for any required preprocessing steps. We will present the details, further analysis and experiments in the full version.

---

[4]We use the same parameters (such as number of queries, etc.) as in the SmartNoise tutorial notebooks. Similarly, for the Adult dataset, we use only the categorical columns as per the notebook [31].

Table 1: Runtime for different values of $T$ (MWEM iterations). Central: Centralized setting runs the MWEM algorithm [31]; Other columns: Distributed setting with 2 data holders and MPC protocols run on different number of computing servers with different security settings: 2PC [8], 3PC [3, 11], 4PC [11]. $|Q|$ is the number of queries, (a x b) denotes the dataset size (dimension).

| DATASET | $T$ | CENTRAL | 2PC PASSIVE | 3PC PASSIVE | 3PC ACTIVE | 4PC ACTIVE |
|---|---|---|---|---|---|---|
| CAR (1,728 X 7) $|Q = 400|$ | 10 | 0.33 SEC | 12.14 SEC | 10.09 SEC | 20.52 SEC | 11.81 SEC |
| | 20 | 0.71 SEC | 23.50 SEC | 20.26 SEC | 43.01 SEC | 23.98 SEC |
| | 30 | 1.30 SEC | 37.86 SEC | 31.91 SEC | 66.40 SEC | 36.22 SEC |
| | 40 | 2.13 SEC | 51.20 SEC | 43.60 SEC | 87.53 SEC | 51.85 SEC |
| ADULT (12,499 X 12) $|Q = 500|$ | 10 | 4.03 SEC | 62.95 SEC | 56.928 SEC | 109.56 SEC | 63.26 SEC |
| | 20 | 4.89 SEC | 152.70 SEC | 147.42 SEC | 229.05 SEC | 127.26 SEC |
| | 30 | 6.38 SEC | 244.80 SEC | 236.10 SEC | 428.87 SEC | 263.54 SEC |
| | 40 | 8.43 SEC | 272.35 SEC | 232.50 SEC | 456.50 SEC | 272.63 SEC |

# 5   Conclusion and Future Work

In this paper we introduced and started the study of a novel approach for generating differentially private synthetic data from distributed databases based on MPC. Our experiments show that utilizing MPC to emulate a central authority produces synthetic datasets with utility at par with data produced in a centralized fashion. While the simplicity of MWEM makes it attractive to many applications and to adapting it to our framework, an important direction for future work is the development of efficient MPC protocols for more recently introduced synthetic data generation techniques such as DPGAN and DP-CGAN [33, 36], drawing inspiration from recent MPC protocols for training of deep neural networks [20].

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

# A Differential Privacy Mechanisms

The exponential mechanism provides the ability of making differentially private selections of the best alternative in a discrete set $\mathcal{K}$, where "best" is based on a scoring function.

**Definition 1** *Exponential Mechanism [23]. Let $s : \mathcal{D} \times \mathcal{K} \to \mathbb{R}$ be a quality scoring function where $s(D, k)$ denotes the quality of result $k$ on dataset $D$ and $\mathcal{K}$ is the set of possible results. The exponential mechanism $E$ selects $k$ from $\mathcal{R}$ such that the probability that a particular $k$ is selected is proportional to $\exp(\epsilon \cdot s(D, k)/2)$. In other words, the exponential mechanism samples $k$ from the distribution satisfying*

$$\Pr[E(D) = k] \propto \exp(\epsilon \cdot s(D, k)/2) \tag{1}$$

To guarantee $\epsilon$-differential privacy, the scoring function $s$ is required to satisfy a stability property, where for each result $k$ the difference $|s(D, k) - s(D', k)|$ is at most the number of records that would have to be added or removed to change $D$ to $D'$.

In the MWEM algorithm, the set of "results" to be selected from at each iteration is the set of queries $Q = \{q_1, q_2, \ldots, q_N\}$, and the value of the scoring function for query $q_i$ is $s(D, q_i) = |q_(A) - q_i(D)|$, i.e. the difference in the answer for query $q_i$ when run over the approximate data $A$ vs. when run over the real data $D$. Alg. 4 provides pseudocode for the exponential mechanism for query selection in-the-clear, i.e. assuming that the data set $D$ has been disclosed in its entirety to a central aggregator [17, 31]. Lines 1–6 generate the probability distribution over the set of results (queries) as per Eq. (1), while lines 8–13 sample a result (query).

---

**Algorithm 4:** Algorithm for sampling a query using the Exponential Mechanism

**Input:** Answers to linear queries for synthetic data $q(A)$ and real data $q(D)$, number of linear queries $N$, and privacy parameter $\epsilon'$.

1   // Compute the probability distribution over the set of queries
2   **for** $i \leftarrow 1$ **to** $N$ **do**
3     $err[i] = 0.5 \cdot \epsilon' \cdot \mathrm{abs}(q_i(A) - q_i(D))$ //Note : $s(D, q_i) = |q_i(A) - q_i(D)|$

4   max_err = max($err$)
5   **for** $i \leftarrow 1$ **to** $N$ **do**
6     $err[i] = \exp(err[i] - $ max_err$)$

7   // Sample the query
8   e_s = $\sum_{i=1}^{N}(err[i])$
9   r = random value drawn from uniform distribution in $[0, 1]$
10   c = 0
11   **for** $i \leftarrow 1$ **to** $N$ **do**
12     c = c + $err[i]$
13     **if** c > r · e_s **then**
       **Output:** return $i$

**Output:** return $N$

---

**Definition 2** $l_1$-*sensitivity. The $l_1$-sensitivty of a function $f : \mathcal{D} \to \mathbb{R}$ is:*

$$\Delta f = \max_{D, D'} \| f(D) - f(D') \|_1 \tag{2}$$

*where $D$ and $D'$ are neighboring databases.*

We note that, for a linear query $f$, $\Delta f = 1$.

The Laplace distribution with $0$ mean and scale $\lambda$, denoted by $\mathsf{Lap}(\lambda)$, has a probability density function $\mathsf{Lap}(x|\lambda) = \frac{1}{2\lambda} e^{-\frac{x}{\lambda}}$. It can be used to obtain an $\epsilon$-differentially private algorithm to answer numeric queries [14].

**Definition 3** *Laplace Mechanism. Let $f : \mathcal{D} \to \mathbb{R}$ be a numeric query. The Laplace mechanism is defined as:*

$$\mathcal{M}_L(x, f(\cdot), \epsilon) = f(x) + \eta \tag{3}$$

*where $\eta$ is drawn from the Laplace distribution $\mathsf{Lap}(\frac{\Delta f}{\epsilon})$.*

# B   Secure Multiparty Computation Primitives

The protocols in Sec. 3 are sufficiently generic to be used in dishonest-majority as well as honest-majority settings, with passive or active adversaries. This is achieved by changing the underlying MPC scheme to align with the desired security setting. In the MPC schemes used in Tab. 1, all computations are done on integers modulo $q$, i.e., in a ring $\mathbb{Z}_q = \{0, 1, \ldots, q-1\}$, with $q$ a power of 2. As is common in MPC, any input real values from the data holders are converted to integers using a fixed-point representation [7]. Below we give a high level description of the 3PC schemes used in Tab. 1. For more details and a description of the other MPC schemes, we refer to the papers about 2PC [8], 3PC [3, 11], and 4PC [11].

**Replicated sharing (3PC).** In a replicated secret sharing scheme with 3 servers (3PC), a value $x$ in $\mathbb{Z}_q$ is secret shared among servers (parties) $S_1, S_2$, and $S_3$ by picking uniformly random shares $x_1, x_2, x_3 \in \mathbb{Z}_q$ such that $x_1 + x_2 + x_3 = x \mod q$, and distributing $(x_1, x_2)$ to $S_1$, $(x_2, x_3)$ to $S_2$, and $(x_3, x_1)$ to $S_3$. Note that no single server can obtain any information about $x$ given its shares. We use $[\![x]\!]$ as a shorthand for a secret sharing of $x$.

**Passive security (3PC).** The 3 servers can perform the following operations through carrying out local computations on their own shares: addition of a constant, addition of secret shared values, and multiplication by a constant. For multiplying secret shared values $[\![x]\!]$ and $[\![y]\!]$, we have that $x \cdot y = (x_1 + x_2 + x_3)(y_1 + y_2 + y_3)$, and so $S_1$ computes $z_1 = x_1 \cdot y_1 + x_1 \cdot y_2 + x_2 \cdot y_1$, $S_2$ computes $z_2 = x_2 \cdot y_2 + x_2 \cdot y_3 + x_3 \cdot y_2$ and $S_3$ computes $z_3 = x_3 \cdot y_3 + x_3 \cdot y_1 + x_1 \cdot y_3$. Next, the servers obtain an additive secret sharing of 0 by picking uniformly random $u_1, u_2, u_3$ such that $u_1 + u_2 + u_3 = 0$, which can be locally done with computational security by using pseudorandom functions, and $S_i$ locally computes $v_i = z_i + u_i$. Finally, $S_1$ sends $v_1$ to $S_3$, $S_2$ sends $v_2$ to $S_1$, and $S_3$ sends $v_3$ to $S_2$, enabling the servers $S_1, S_2$ and $S_3$ to get the replicated secret shares $(v_1, v_2)$, $(v_2, v_3)$, and $(v_3, v_1)$, respectively, of the value $v = x \cdot y$. This protocol only requires each server to send a single ring element to one other server, and no expensive public-key encryption operations (such as homomorphic encryption or oblivious transfer) are required. This MPC scheme was introduced by Araki et al. [3].

**Active security (3PC).** In the case of malicious adversaries, the servers are prevented from deviating from the protocol and gain knowledge from another party through the use of information-theoretic message authentication codes (MACs). For every secret share, an authentication message is also sent to authenticate that each share has not been tampered in each communication between parties. In addition to computations over secret shares of the data, the servers also need to update the MACs appropriately, and the operations are more involved than in the passive security setting. For each multiplication of secret shared values, the total amount of communication between the parties is greater than in the passive case. We use the MPC scheme `SPDZ-wiseReplicated2k` recently proposed by Dalskov et al. [11] that is available in MP-SPDZ [19].

**MPC primitives.** The MPC schemes listed above provide a mechanism for the servers to perform cryptographic primitives through the use of secret shares, namely addition of a constant, multiplication by a constant, and addition of secret shared values, and multiplication of secret shared values (denoted as $\pi_{\mathsf{MUL}}$). Building on these cryptographic primitives, MPC protocols for other operations have been developed in the literature. We use [19]:

- Secure random number generation from uniform distribution $\pi_{\mathsf{GR-RANDOM}}$ : In $\pi_{\mathsf{GR-RANDOM}}$, each party generates $l$ random bits, where $l$ is the fractional precision of the power 2 ring representation of real numbers, and then the parties define the bitwise XOR of these $l$ bits as the binary representation of the random number jointly generated.
- Secure random bit generation $\pi_{\mathsf{GR-RNDM-BIT}}$ : In $\pi_{\mathsf{GR-RNDM-BIT}}$, each party generates the secret share of a single random bit, such that the generated bit is either 0 or 1 with a probability of 0.5.
- Secure equality test $\pi_{\mathsf{EQ}}$ : At the start of this protocol, the parties have secret sharings $[\![x]\!]$; at the end if $x = 0$, then they have a secret share of 1, else a secret sharing of 0.
- Secure less than test $\pi_{\mathsf{LT}}$ : At the start of this protocol, the parties have secret sharings $[\![x]\!]$ and $[\![y]\!]$ of integers $x$ and $y$; at the end of the protocol they have a secret sharing of 1 if $x < y$, and a secret sharing of 0 otherwise.
- Secure greater than test $\pi_{\mathsf{GT}}$ : At the start of this protocol, the parties have secret sharings $[\![x]\!]$ and $[\![y]\!]$ of integers $x$ and $y$; at the end of the protocol they have a secret sharing of 1 if $x > y$, and a secret sharing of 0 otherwise.

- Other primitives : We use secure maximum protocol ($\pi_{\mathsf{MAX}}$), secure exponential protocol ($\pi_{\mathsf{EXP}}$) and secure logarithm protocol ($\pi_{\mathsf{LN}}$) as the building blocks for our protocols. $\pi_{\mathsf{LN}}$ uses the polynomial expansion for computing logarithm and $\pi_{\mathsf{EXP}}$ in turn uses the $\pi_{\mathsf{LN}}$ to compute exponential. $\pi_{\mathsf{MAX}}$ inherently uses the $\pi_{\mathsf{GT}}$ repeatedly over a list by employing variant of Divide-n-Conquer approach. At the start of all of these primitives, parties hold the secret sharings $[\![x]\!]$ and at the end of the protocol they hold the secret shares of the corresponding computed values.

MPC protocols can be mathematically proven to guarantee privacy and correctness. We follow the universal composition theorem that allows modular design where the protocols remain secure even if composed with other or the same MPC protocols [6].

**Implementing DP in MPC.** Keeping in mind the dangers of implementing DP with floating point arithmetic [24], we stick with the best practice of using fixed-point and integer arithmetic as recommended by, for example, OpenDP [5]. We implement all our DP mechanism using their discrete representations and use 32 bit precision to ensure correctness.

We also remark that finite precision issues can also impact the exponential mechanism. If the utility of one of the classes in the exponential mechanism collapses to zero after the mapping into finite precision, pure DP becomes impossible to achieve. We can deal with such situation by using approximate DP for an appropriate value of $\delta$. Such situation did not happen with the data sets used in our experiments.

---

[5]https://opendp.org/

