# OpenReview forum: "Secure Multiparty Computation for Synthetic Data Generation from Distributed Data"
_NeurIPS.cc/2022/Workshop/SyntheticData4ML — Neurips 2022 SyntheticData4ML_

### Official Review · Reviewer_Trz7 · 2022-10-17
**Secure Multiparty Computation for Synthetic Data Generation from Distributed Data**

**Rating:** 6
**Confidence:** 5

**Review:**

MWEM algorithm is a popular technique to release differential private datasets in a centralized setting.
The paper tried to run the MWEM algorithm through MPC protocols for generating synthetic data under
differential privacy guarantees in a decentralized setting. The paper also conducted experiments on two datasets to
compare the results (through AUC-ROC metric) between the centralized and proposed approach.

When I read the experimental results from Table 1, it is strange to see that the
4PC – active values for CAR dataset are smaller than the 3PC active values of CAR datasets.
In the case of ADULT datasets, the 4PC active values are greater than the 3PC active values.
There is no explanation available in the paper for this strange behavior between the two datasets.

The MPC protocol security assumption related to whether the adversary is static
(Corrupted before the protocol begins) or adaptive (corrupted during the execution of the protocol)
is also missing in the paper.

Even though the proposed approach is very interesting, one of the fundamental problems
with the MPC protocols is that the communication complexity increases exponentially
with the increase in the number of players participating in the protocol.
There are efforts made to scale the MPC protocols with respect to the number of players
however, all these efforts work under specific assumptions.
The paper has not introduced any such scalability approaches to minimize the
communication complexity between increasing number of players participating in the protocol.

Furthermore, another major drawback with the naïve MPC protocol-based approach is that all the players (computing parties)
participating in the MPC protocol need to be online and interactive during the execution of the protocol, which may not be very realistic in large scale settings.

In essence, a naïve MPC approach as proposed in the paper to decentralize the synthetic data generation process under differential
privacy may work for a small number of players under very strict assumptions but do not scale well and
can’t be used as a generic approach without having additional blocks to minimize the communication complexity
and to address other security guarantees (addressing adaptive adversary, man-in-the middle, and then supply-chain attacks, accountability, etc).

Despite all the weaknesses in the paper, the topic of the paper and the solution direction
has a potential to trigger a lot of interesting discussions about generating synthetic data in decentralized settings
during the workshop.

---

### Official Review · Reviewer_RUvN · 2022-10-18
**Beautiful MPC on a useless synthetic data generation model**

**Rating:** 6
**Confidence:** 4

**Review:**

The paper presents a beautiful and well-thought MPC approach for applying DP synthetic data generation, but does this based on a completely antiquated MWEM model with very poor performance.

In the future, I hope the authors would apply their MPC expertise to state-of-the-art DP synthetic data generation. I would especially recommend the authors to check more recent graphical-model-based approaches, such as PGM (McKenna et al., 2019) as well as other recent work. The PGM paper shows how their approach very significantly improves over MWEM.

The paper includes a discussion of potential weaknesses to side-channel attacks and concerns with DP implemented on floating point. It does, however, miss a major weakness in exponential mechanism with finite precision.

Specifically, if the utility of one of the options is so poor under one of the possible data sets that its probability underflows to zero, the proposed sampling scheme will never pick that option and the algorithm will not be epsilon-DP with any finite $\epsilon$. This can be addressed by relaxing to approximate $(\epsilon, \delta)$-DP.

Still, the discussion of the MPC methods seems useful, although the specific method proposed here has zero practical utility, as the classification accuracy on Adult is close to random guessing level even with $\epsilon=50$, while current state-of-the-art can reach close to non-private performance with much smaller $\epsilon$ (although in approximate DP, but then again the proposed method only satisfies approximate DP as well).

---

### Official Review · Reviewer_TmF6 · 2022-10-18
**This is an interesting paper applying secure multiparty computation for the generation synthetic data using a modified algorithm, i.e. Multiplicative Weights with Exponential Mechanism**

**Rating:** 6
**Confidence:** 3

**Review:**

In the paper «Secure Multiparty Computation for Synthetic Data Generation from Distributed Data» the authors propose to tackle problems with sharing/analyzing data without exposing data between entities taking part in the analysis, or a central aggregator. They propose a solution using secure multiparty computation. As a use-case the present generation of synthetic data and conduct experiments on two datasets compare the results (through AUC-ROC metric) between the centralized and proposed approach.

The novelty of the paper is the use the MPC based protocol for synthetizing data, while the generation of synthetic data is done by using a well-known and used algorithm, but with modification for providing differential privacy, on two open datasets that has previously been used for generating synthetic data.

A well-known drawback of the MPC protocols is that the communication complexity increases exponentially with the increase of number of data holders/computing parties.
The paper should address when in which situations the MPC approach is feasible with regard to number of participants.

Even though the data set are openly available, they are not known to all, and it would be beneficial if the authors briefly described size of the dataset in relation to runtime and scalability as the dataset are rather small, as well what kind/types of data the proposed system is able to handle.

---

### Meta-Review · Area_Chair_EUxr · 2022-10-19

**Recommendation:** Accept